# Pharmacotherapy, drug-drug interactions and potentially inappropriate medication in depressive disorders

Jan Wolff[1,2,3]*, Pamela Reißner[4], Gudrun Hefner[5], Claus Normann[2], Klaus Kaier[6], Harald Binder[6], Christoph Hiemke[7], Sermin Toto[8], Katharina Domschke[2], Michael Marschollek[1], Ansgar Klimke[4,9]

1 Peter L. Reichertz Institute for Medical Informatics of TU Braunschweig and Hannover Medical School, Hannover, Germany, 2 Faculty of Medicine, Department of Psychiatry and Psychotherapy, Medical Center - University of Freiburg, University of Freiburg, Freiburg, Germany, 3 Evangelical Foundation Neuerkerode, Braunschweig, Germany, 4 Vitos Hochtaunus, Friedrichsdorf, Germany, 5 Vitos Clinic for Forensic Psychiatry, Eltville, Germany, 6 Faculty of Medicine, Institute of Medical Biometry and Statistics, Medical Center - University of Freiburg, University of Freiburg, Freiburg, Germany, 7 Department of Psychiatry and Psychotherapy, University Medical Center Mainz, Mainz, Germany, 8 Department of Psychiatry, Social Psychiatry and Psychotherapy, Hannover Medical School, Hannover, Germany, 9 Heinrich-Heine-University Düsseldorf, Düsseldorf, Germany

* janwolff123@gmail.com, wolff.jan@mh-hannover.de

**Data Availability Statement:** Our study required potentially identifying and sensitive human research participant data, such as psychiatric diagnoses, medication data and many others. We

## Abstract

### Introduction

The aim of this study was to describe the number and type of drugs used to treat depressive disorders in inpatient psychiatry and to analyse the determinants of potential drug-drug interactions (pDDI) and potentially inappropriate medication (PIM).

### Methods

Our study was part of a larger pharmacovigilance project funded by the German Innovation Funds. It included all inpatients with a main diagnosis in the group of depressive episodes (F32, ICD-10) or recurrent depressive disorders (F33) discharged from eight psychiatric hospitals in Germany between 1 October 2017 and 30 September 2018 or between 1 January and 31 December 2019.

### Results

The study included 14,418 inpatient cases. The mean number of drugs per day was 3.7 (psychotropic drugs = 1.7; others = 2.0). Thirty-one percent of cases received at least five drugs simultaneously (polypharmacy). Almost one half of all cases received a combination of multiple antidepressant drugs (24.8%, 95% CI 24.1%–25.5%) or a treatment with antidepressant drugs augmented by antipsychotic drugs (21.9%, 95% CI 21.3%–22.6%). The most frequently used antidepressants were selective serotonin reuptake inhibitors, followed by serotonin and norepinephrine reuptake inhibitors and tetracyclic antidepressants. In multivariate analyses, cases with recurrent depressive disorders and cases with severe depression were more likely to receive a combination of multiple antidepressant drugs (Odds ratio

cannot share these data because legal restrictions in Germany and the European Union ("General Data Protection Regulation" and "Processing of special categories of personal data") prohibit sharing these sensitive data. Further information can be obtained from the ethics committee of the State Medical Association of Hesse, info@laekh.de, Hanauer Landstraße 152, 60314 Frankfurt, Germany, under the file number FF116 / 2017."

**Funding:** Financial support for this study was provided by a grant from the Innovations Funds of the German Federal Joint Committee (grant number: 01VSF16009, www.g-ba.de/english/). The funders had no role in study design, data collection and analysis, decision to publish, or preparation of the manuscript.

**Competing interests:** Independent of the present study, KD received fees from Janssen Pharmaceuticals, Inc. for her consultancy work on the Neuroscience Steering Committee. CN received lecture and consultancy fees from Janssen-Cilag and Neuraxpharm as well as fees for conducting clinical studies from Janssen-Cilag. ST has received lecture fees from Janssen-Cilag, Otsuka / Lundbeck and Servier and is a member of the Advisory Board of Otsuka and Janssen-Cilag. CH has received lecture fees from Otsuka. The commercial funding sources were unrelated to the present study and did not alter our adherence to PLOS ONE policies on sharing data and materials.

recurrent depressive disorder: 1.56, 95% CI 1.41–1.70, severe depression 1.33, 95% CI 1.18–1.48). The risk of any pDDI and PIM in elderly patients increased substantially with each additional drug (Odds Ratio: pDDI 1.32, 95% CI: 1.27–1.38, PIM 1.18, 95% CI: 1.14–1.22) and severity of disease (Odds Ratio per point on CGI-Scale: pDDI 1.29, 95% CI: 1.11–1.46, PIM 1.27, 95% CI: 1.11–1.44), respectively.

## Conclusion

This study identified potential sources and determinants of safety risks in pharmacotherapy of depressive disorders and provided additional data which were previously unavailable. Most inpatients with depressive disorders receive multiple psychotropic and non-psychotropic drugs and pDDI and PIM are relatively frequent. Patients with a high number of different drugs must be intensively monitored in the management of their individual drug-related risk-benefit profiles.

## Introduction

Depressive disorders were the third leading cause of global non-fatal burden of disease in 2017 [1]. Pharmacotherapy is an important component in the treatment of depressive disorders [2]. Common guidelines recommend monotherapy with second generations antidepressants, i.e. selective serotonin reuptake inhibitors (SSRIs), serotonin and norepinephrine reuptake inhibitors (SNRIs) and other drugs that selectively target neurotransmitters [3–5]. The majority of patients fail to achieve remission after first monotherapy with antidepressants [6]. Several second-step treatments are recommended in guidelines, such as switching to a different monotherapy, augmentation with antipsychotics or combining two antidepressants [4,5,7].

The combination of multiple antidepressants and the combination with other drugs bear the risk of potential drug-drug-Interactions (pDDI). pDDI are a frequent cause of adverse drug reactions (ADR) [8]. The number of simultaneously taken drugs is one of the strongest risk factors for pDDI and potentially inappropriate medication in the elderly (PIM) [9]. pDDI are a relevant aspect in the treatment of patients with depressive disorders, for instance, via metabolism by the cytochrome P450 enzyme group (pharmacokinetic) and the combination of multiple anticholinergics or QT-interval prolonging drugs (pharmacodynamic).

These pDDI are of specific relevance for the inpatient treatment of depressive disorders. Cytochrome P450 (CYP) enzymes are essential for the phase 1 metabolism of drugs and most pharmacokinetic pDDI in the treatment of depressive disorders are the results of inhibition or induction of CYP enzymes [10]. Many drugs for the treatment of depressive disorders have strong anticholinergic effects in connection with their biochemical mechanisms, such as tricyclic antidepressants [11,12]. Drug-induced prolongation of the QT interval is associated with an increased risk of a rare but potentially fatal form of cardiac arrhythmia, so-called "torsade de pointes" (TdP), [13]. A prolongation of the QT interval has been shown for several antidepressants, in particular tricyclic antidepressants and the SSRIs (es-) citalopram [14,15]. An increased risk of ADR was found with the simultaneous use of more than one anticholinergic [16] and QT interval prolonging drug [17].

Pharmacokinetics and -dynamics change in elderly patients due to the progressive decline in the functional reserve of multiple organs and systems with an influence on drug disposition when aging [18]. Medication is considered as PIM if risks outweigh benefits of better

alternatives [19–21]. Drug safety requires that these aspects are taken into account in the treatment of depressive disorders in elderly patients.

The pharmacological treatment of depressive disorders is complex and the associated pDDI and PIM can impair patient outcomes and increase costs [22]. The aim of this study was to describe the number and type of drugs used to treat depressive disorders in inpatient psychiatry and to analyse the patient- and treatment-specific determinants of pDDI and PIM.

## Methods

### Study sample

Our study included all inpatient cases with a main diagnosis in the group of depressive episodes (F32*, International statistical classification of diseases and related health problems.–10th revision, ICD-10) or recurrent depressive disorders (F33*) who were consecutively discharged from eight psychiatric hospitals between 1 October 2017 and 30 September 2018 or between 1 January and 31 December 2019. These hospitals belong to a common health care provider that provides about one half of all inpatient psychiatric services in Hesse, Germany. The present study was part of a larger pharmacovigilance project funded by the German Innovation Funds (OSA-PSY—Optimization of inpatient drug therapy for mental illnesses, grant number 01VSF16009). The German Innovation Funds sponsors innovative projects to improve the quality of medical care provided under the statutory health insurance system. The aim of the larger project was to use daily patient-specific medication data and their dissemination among clinical staff to improve drug safety in inpatient psychiatry. The study was approved by the ethics committee of the State Medical Association of Hesse under the file number FF116/2017. In accordance with the ethics approval, our retrospective study did not require individual patient consent. The present study analysed a sub-sample of the total research project, namely patients with depressive disorders, i.e. a main diagnosis of F32* or F33*, ICD-10. Previous publications from this research project can be found in the reference list [10,12,14,23–26].

### Medication data

We used daily medication data for each included inpatient obtained from the electronic medical records at the study sites. Thereby, we were able to investigate the medications for each day separately and to include all treatment modifications of during a hospital stay. The pDDI analysed by our study were defined as 1) pharmacokinetic pDDI via CYP enzyme inducing and inhibiting drugs and the respective victim drugs (CYP450-Interaction), 2) pharmacodynamic pDDI via the administration of more than one anticholinergic drug (Antichol.-Combi.) and 3) pharmacodynamic pDDI via administration of more than one drug that potentially prolongs the QT-interval (QT-Combi.). In addition, the administration of PIM to patients over the age of 64 years was investigated.

CYP-mediated drugs were identified in accordance to the Consensus Guidelines for Therapeutic Drug Monitoring in Neuropsychopharmacology [27], restricted to inhibitions and inductions that lead to decrease or increase of plasma concentrations of victim drugs by more than 50%, respectively. In addition, melperone [28], levomepromazine [29] and perazine [30–32] were considered as CYP inhibitors. Additional non-psychotropic victim drugs were added based on CYP substrate properties defined by Hiemke and Eckermann [33]. In total, these sources resulted in covering the following isoforms for analyses of CYP-mediated pDDI: CYP1A2, CYP2B6, CYP2C19, CYP2C9, CYP2D6, CYP2E1, CYP3A4.

QT interval prolonging drugs were identified based on the lists of Hiemke and Eckermann [33], Wenzel-Seifert and Wittmann [15] and the drugs listed with known or possible risk for

TdP by Arizona Center for Education and Research on Therapeutics (AZCERT) [34,35]. Anticholinergic activity of drugs was identified according to Hiemke and Eckermann, Chew et al. and Lertxundi et al. [33,36,37]. PIM were identified according the German list of medications that are potentially inappropriate in elderly patients, the so-called Priscus-list [19].

The groups N05 and N06 of the Anatomical-Therapeutic-Chemical (ATC) classification, respectively, were defined as psychotropic drugs [38]. Drugs classified in group N06A were defined as antidepressants. Drugs classified in group N05A were defined as antipsychotics. Dietetics and food supplements, homeopathic preparations and anthroposophic medicine and only locally applied active ingredients were excluded. We defined polypharmacy as the simultaneous use of at least five different pharmaceuticals [39], averaging over the entire hospital stay.

We differentiated antidepressant drug regimens between a) monotherapy, i.e. receiving one antidepressant drug, b) switch/trial, i.e. receiving more than one antidepressant or antipsychotic drug but not more than three days in combination, c) antidepressant combination, i.e. receiving more than one antidepressant drug in combination more than three days and d) augmentation, i.e. receiving a combination of antidepressant and antipsychotic drugs more than three days [40].

### Analyses and measurements

We obtained patient and treatment data from the patient administration databases of each treatment site. These data were patient gender, age at admission, length of stay, treatment type (i.e. day-clinic versus regular ward), the Clinical Global Impressions at admission [41] and main diagnoses and all psychiatric and somatic comorbidities according to the ICD-10. These data were used to describe the study sample and to adjust for potential confounders in multivariate models.

Arithmetic means and standard deviations were calculated as measures of central tendency and dispersion, respectively. Medians and interquartile ranges were calculated for variables with skewed distributions or a relevant number of outliers. Confidence intervals for proportions were calculated according to Agresti and Coull [42]. We used multivariate logistic regression models to explain the relationship between patient-specific characteristics and type of antidepressant treatment and the outcome of at least one pDDI and at least on PIM during the hospital stay, respectively.

### Results

The study included 14,418 inpatient cases from eight psychiatric hospitals (Table 1). About 61% of total cases had a main diagnosis in the group recurrent depressive disorders (F33*) and 39% in the group of depressive episodes (F32*). Thirty-one percent of cases received at least 5 drugs simultaneously (polypharmacy). A total of 96 different psychotropic and 619 different non-psychotropic drugs were administered during the study period. The mean daily number of drugs was 3.7 (psychotropic drugs = 1.7; others = 2.0) per case.

Antidepressants were by far the most frequently used drug group, with 85% of cases with recurrent depressive disorders and 73% of depressive episodes receiving at least one drug from this group. The second most frequently used group were antipsychotic drugs, which were used for 61% of cases with recurrent depressive disorders and 52% of depressive episodes.

Fig 1 shows how frequent and in which combinations antidepressant and antipsychotic drugs were used. Almost one half of all cases received a combination of multiple antidepressant drugs (24.8%, 95% CI 24.1%–25.5%) or a treatment with antidepressant drugs augmented by antipsychotic drugs (21.9%, 95% CI 21.3%–22.6%). The most frequently used

**Table 1. Description of included cases.**

| | | |
|---|---|---|
| Number of cases | 14,418 | |
| Female (Number, % of total per column) | 8,307 | 58 |
| Age at admission (in years, mean and standard deviation) | 48 | 18 |
| Length of stay (in days, median and interquartile range) | 29 | 14 to 46 |
| Day-clinic (Number of cases, % of total per column) | 4,159 | 29 |
| Number of comorbidities (median and interquartile range) | 2 | 1 to 3 |
| Main diagnosis (Number of cases, % of total per column) | | |
| (F32.0) Mild depressive episode | 14 | 0 |
| (F32.1) Moderate depressive episode | 1,573 | 11 |
| (F32.2) Severe depressive episode without psychotic symptoms | 3,538 | 25 |
| (F32.3) Severe depressive episode with psychotic symptoms | 475 | 3 |
| (F32.8) Other depressive episodes | 22 | 0 |
| (F32.9) Depressive episode, unspecified | 22 | 0 |
| (F33.0) Recurrent depressive disorder, current episode mild | 21 | 0 |
| (F33.1) Recurrent depressive disorder, current episode moderate | 2,509 | 17 |
| (F33.2) Recurrent depressive disorder current episode severe without psychotic symptoms | 5,613 | 39 |
| (F33.3) Recurrent depressive disorder current episode severe with psychotic symptoms | 605 | 4 |
| (F33.4) Recurrent depressive disorder, currently in remission | 13 | 0 |
| (F33.8) Other recurrent depressive disorders | 7 | 0 |
| (F33.9) Recurrent depressive disorder, unspecified | 6 | 0 |
| Number of psychotropic drugs per day (mean and standard deviation) | 1.7 | 1.1 |
| Number of non-psychotropic drugs per day (mean and standard deviation) | 2.0 | 2.5 |
| Polypharmacy (Number, % of total per column) | 4,413 | 31 |

Interquartile range shows the values of the 25th and 75th percentiles. Polypharmacy: At least five different drugs simultaneously.

antidepressants were selective serotonin reuptake inhibitors (SSRI), followed by serotonin and norepinephrine reuptake inhibitors (SNRI) and tetracyclic antidepressants (Fig 1B). SSRI were the drug class which was most frequently used for monotherapy (12% of inpatient days, Fig 1A). The most frequent augmentation therapy was SSRI augmented by atypical antipsychotics, followed by SNRI augmented by atypical antipsychotics. The most frequent combinations of antidepressants were SSRI in combination with tetracyclic antidepressants and SNRI in combination with the tetracyclic antidepressant mirtazapine. Fig 1 shows only drug combinations that accounted for at least 2% of total patient days. Lithium did not reach that threshold. In total, 2.8% of patients received Lithium alone or in combination with other drugs for at least one day.

Table 2 shows the determinants of receiving different antidepressant and antipsychotic drug regimes. Cases with moderate depression and cases with depressive episodes (F32) were more likely to receive neither antidepressant nor antipsychotic drug treatment and more likely to receive a monotherapy with antidepressants. Cases with recurrent depressive disorders and cases with severe depression were more likely to receive a combination of multiple antidepressant drugs. Cases with a high severity of disease measured by the CGI-scale at both admission and discharge were more likely to have switched between different antidepressant or antipsychotic drugs.

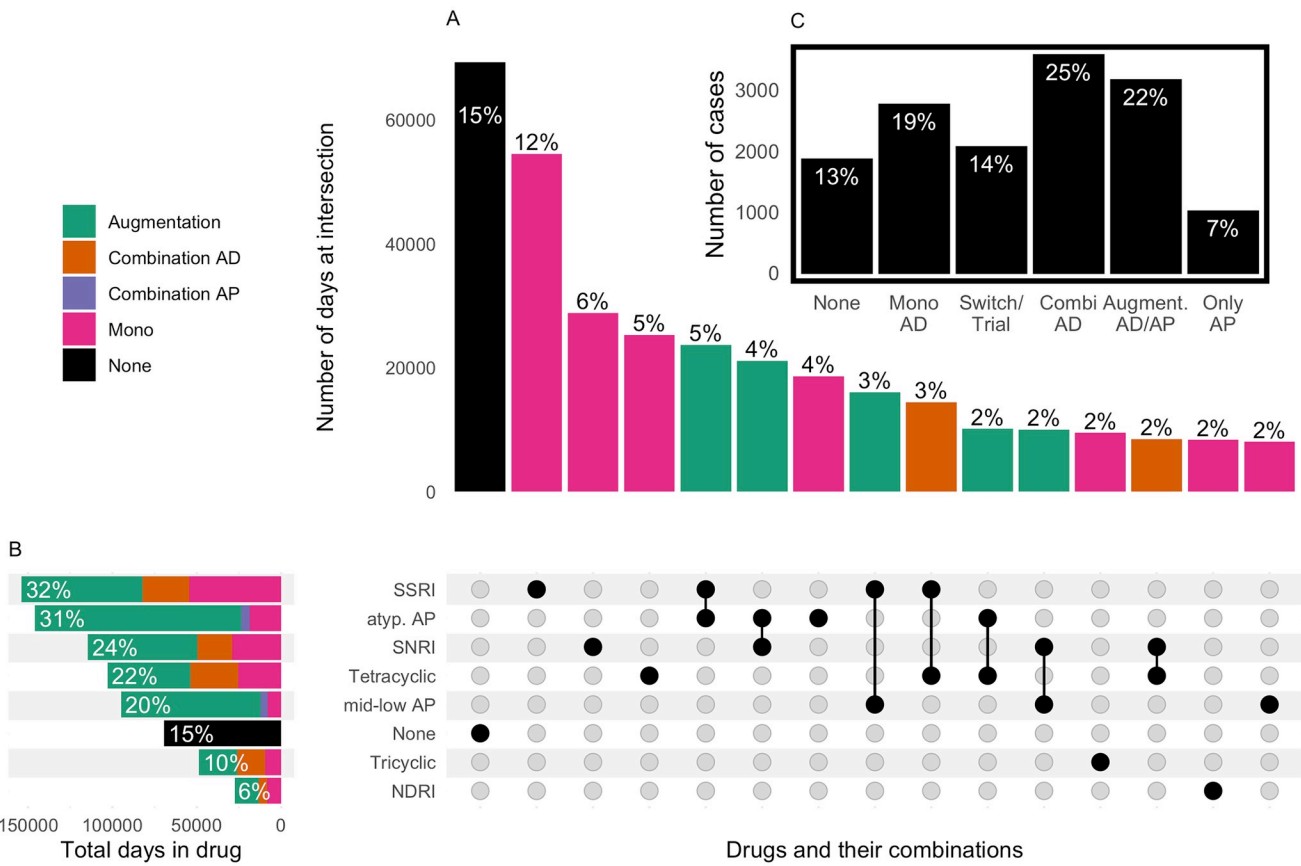

**Fig 1. Frequency of antidepressant and antipsychotic drugs and their combinations. (A+B)** Augmentation: Combination of antidepressant and antipsychotic drugs. Combination AD: Combination of more than one antidepressant drug. Combination AP: Combination of more than one antipsychotic drug. Mono: Use of a single antidepressant or antipsychotic drug. None: Neither an antidepressant drug nor an antipsychotic drug. **(A)** Intersection: Number of days with the respective drug or drug combination and the proportion in total patient days. For instance, at 5% of all inpatient days a drug from the class of SSRIs was augmented by a drug from the class of atypical antipsychotics (atyp. AP). Only intersections with at last 2% of total days are shown. SSRI: Selective serotonin reuptake inhibitors, atyp. AP: Atypical antipsychotics, SNRI: Serotonin and norepinephrine reuptake inhibitors, mid-low AP: middle- and low-potency antipsychotics. None: Patient did not receive an antidepressant or an antipsychotic drug at that day. NDRI: Norepinephrine–dopamine reuptake inhibitor **(B)** Total number of days with the respective drug and the proportion in total patient days. Double counting of days is possible. Combinations are counted within and between classes of antidepressants and antipsychotics. **(C)** Total number of cases with the respective treatment regime and proportion in total cases. None: Patient did not receive an antidepressant or an antipsychotic drug. Mono AD: Patient received one antidepressant drug. Switch/Trial: Patient received more than one antidepressant or antipsychotic drug but not more than three days in combination. Combi AD: Patient received more than one antidepressant drug in combination more than three days. Augmentation AD/AP: Patient received a combination of antidepressant and antipsychotic drugs more than three days. Only AP: Patient received one or more antipsychotic drugs but no antidepressant drug.

The number of simultaneously used drugs influenced the risk of pDDI and PIM, as illustrated in Fig 2. The steepest increase and the highest overall risk were found in pharmacodynamic pDDI related to the combination of multiple QT-prolonging drugs.

Fig 3 shows the TOP-20 drugs and drug combinations of each field of pDDI and PIM, respectively. The three most frequently in CYP450-Interactions involved single drugs were Duloxetine, Melperone and Bupropion, accounting for 30%, 21% and 17% of all cases affected by CYP450-Interactions, respectively. The three most frequently Antichol.-Combi. involved single drugs were Promethazine (49%), Olanzapine (40%) and Amitriptyline (28%). The three most frequently in QT-Combi. involved single drugs were Mirtazapine (42%), Quetiapine (34%) and Pipamperone (28%).

**Table 2. Logistic regression models on the probability of receiving different antidepressant or and antipsychotic drug treatments.**

| | None | | Mono AD | | Switch/Trial | | Combi AD | | Only AP | | Augmentation AD/AP | |
|---|---|---|---|---|---|---|---|---|---|---|---|---|
| | Odds Ratio | 95% CI | Odds Ratio | 95% CI | Odds Ratio | 95% CI | Odds Ratio | 95% CI | Odds Ratio | 95% CI | Odds Ratio | 95% CI |
| Day clinic | 2.10 | 1.81 2.39 | 1.96 | 1.75 2.18 | 0.61 | 0.33 0.89 | 0.87 | 0.77 0.96 | 0.62 | 0.50 0.74 | 1.11 | 0.73 1.48 |
| Sex (female) | 0.95 | 0.84 1.07 | 1.05 | 0.95 1.15 | 1.09 | 0.82 1.36 | 1.09 | 0.99 1.19 | 0.83 | 0.71 0.96 | 0.96 | 0.76 1.16 |
| Age (10 y.) | 0.78 | 0.75 0.81 | 0.98 | 0.95 1.01 | 1.07 | 0.89 1.24 | 1.22 | 1.19 1.25 | 0.80 | 0.76 0.83 | 0.95 | 0.81 1.09 |
| *Main diag. (Ref: F32)* | | | | | | | | | | | | |
| (F33) Recurrent depr. dis. | 0.52 | 0.46 0.58 | 0.81 | 0.73 0.89 | 1.16 | 0.88 1.44 | 1.56 | 1.41 1.70 | 0.83 | 0.71 0.95 | 0.86 | 0.69 1.04 |
| *Severity (Ref: Moderate)* | | | | | | | | | | | | |
| Severe | 0.52 | 0.45 0.58 | 0.74 | 0.66 0.82 | 0.95 | 0.64 1.25 | 1.33 | 1.18 1.48 | 1.03 | 0.85 1.21 | 1.56 | 1.11 2.00 |
| Mild/Other | 1.00 | 0.40 1.60 | 0.90 | 0.42 1.39 | 0.49 | 0.00 1.22 | 0.83 | 0.33 1.34 | 1.62 | 0.44 2.80 | 1.61 | 0.21 3.00 |
| Number of comorbidities | 0.91 | 0.88 0.94 | 0.93 | 0.90 0.95 | 0.99 | 0.96 1.03 | 0.99 | 0.97 1.01 | 1.11 | 1.08 1.14 | 1.02 | 0.99 1.05 |
| CGI admission | 0.87 | 0.80 0.95 | 0.92 | 0.85 0.98 | 1.25 | 1.03 1.47 | 1.02 | 0.95 1.09 | 1.03 | 0.92 1.14 | 1.08 | 0.92 1.23 |
| CGI discharge | 0.88 | 0.83 0.94 | 0.94 | 0.90 0.99 | 1.17 | 1.05 1.29 | 1.10 | 1.06 1.15 | 0.97 | 0.90 1.04 | 0.98 | 0.90 1.07 |
| Length of stay | 0.97 | 0.96 0.97 | 0.99 | 0.99 0.99 | 0.99 | 0.99 1.00 | 1.02 | 1.02 1.02 | 0.98 | 0.98 0.99 | 1.00 | 1.00 1.00 |

None: Patient did not receive an antidepressant or an antipsychotic drug. Mono AD: Patient received one antidepressant drug. Switch/Trial: Patient received more than one antidepressant or antipsychotic drug but not more than three days in combination. Combi AD: Patient received more than one antidepressant drug in combination therapy more than three days. Only AP: Patient received one or more antipsychotic drugs but no antidepressant drug. Augmentation AD/AP: Patient received a combination of antidepressant and antipsychotic drugs more than three days. CI: Confidence interval. Age (10 y.): Coefficient per ten years of age. Recurrent depr. dis.: Recurrent depressive disorder, CGI: Clinical Global Impression Scale

Fig 4 shows the results of a logistic regression on the occurrence of at least one pDDI and PIM during a patient stay at the hospital. The odds ratios shown in the circles reflect the multiplicative effect, e.g. the risk for receiving at least one PIM increased by 18% for each additional drug taken, controlled for the other variables in the model. Another important risk factor was severity of disease, represented by the admission score on the Clinical Global Impressions (CGI)-Scale. The risk of any pDDI increased with each additional drug by 32% (Odds Ratio: 1.32, 95% CI: 1.27–1.38) and with each additional point at the CGI-scale by 29% (Odds Ratio: 1.29, 95% CI: 1.11–1.46) (not shown in Fig 4).

## Discussion

### Main findings

This study described the number and type of drugs used to treat depressive disorders in inpatient psychiatry and analysed the determinants of pDDI and PIM. Almost one half of all cases received a combination of multiple antidepressant drugs or a treatment with antidepressant drugs augmented by antipsychotic drugs. Cases with recurrent depressive disorders and cases with severe depression were more likely to receive a combination of multiple antidepressant drugs. pDDI and PIM were frequent in patients with depressive disorders, and the main risk factors were the number of simultaneously taken drugs and severity of disease. Relatively few drugs accounted for a large proportion of total pDDI and PIM.

### Clinical implications of different antidepressant drug regimes

Psychiatric hospital care must focus on pharmacovigilance due to patients' frequent exposure to long-term poly-pharmacotherapy, poor compliance to pharmacological treatment and

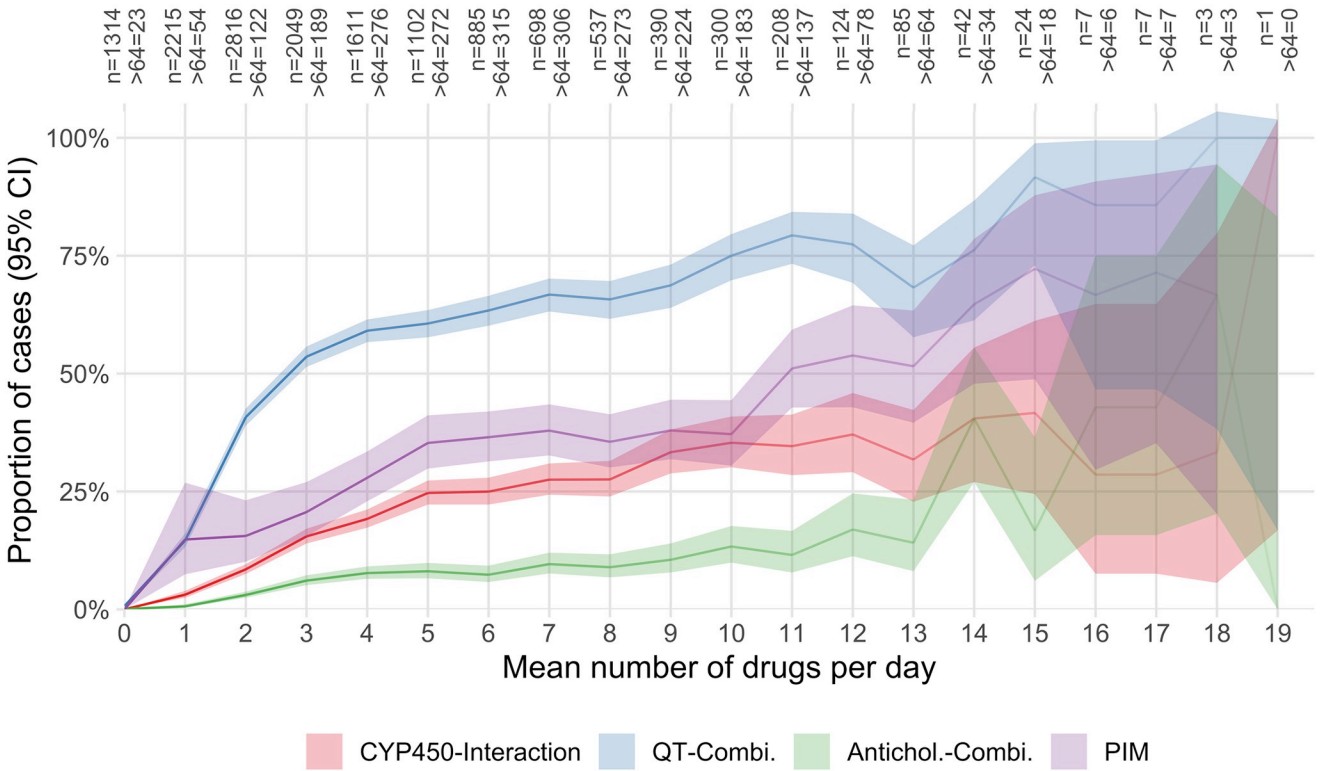

**Fig 2. Mean number of drugs per day and the proportion of cases with at least one pDDI or PIM during their stay.** CYP450-Interaction: Pharmacokinetic cytochrome P450 (CYP)-mediated drug-drug interaction. QT-Combi.: A combination of at least two drugs on the same day with known or possible risk of TdP. Antichol. Combi.: A combination of at least two drugs on the same day with at least moderate anticholinergic activity. PIM: Potentially inappropriate medication in the elderly. >64: Number of cases of more than 64 years of age at admission.

co-morbidity with organic illnesses requiring the prescription of multiple drugs [43]. Pharmacoepidemiologic data about prescription patterns can help to clarify potential sources of safety risks and focus on relevant aspects of drug treatment [44]. This study has added important data on this issue that were previously unavailable.

Antidepressant drugs were the most frequently used pharmacotherapy in our study. The effectiveness of antidepressants in the treatment of depressive disorders has been firmly established by previous research. A recent review investigated more than 500 trials and concluded that all of 21 studied antidepressants were more efficacious than placebo in adults with major depressive disorders [45]. Accordingly, antidepressant monotherapy is recommended as first line therapy for patients with a diagnosis of depressive disorders in current guidelines [5].

The present study found that 13% of cases with depressive disorders neither received antidepressant nor antipsychotic drugs. Cases with moderate depression and cases with depressive episodes were more likely to receive neither antidepressants nor antipsychotic drugs than patients with severe depression and recurrent depressive disorders, respectively. Guidelines recommend antidepressants in exceptional cases for patients with mild depression, generally for moderate depression and especially for patients with severe depression [46]. However, while the benefit of antidepressants over placebo was found to be substantial in severe depression, effects may be minimal or nonexistent, on average, in patients with mild or moderate symptoms [47,48]. In addition to less severe symptoms, there are other potential reasons for a hospital treatment without antidepressants, such as patients' refusal to take the medication [49]. When antidepressant treatment is not indicated or possible or failed, several alternative

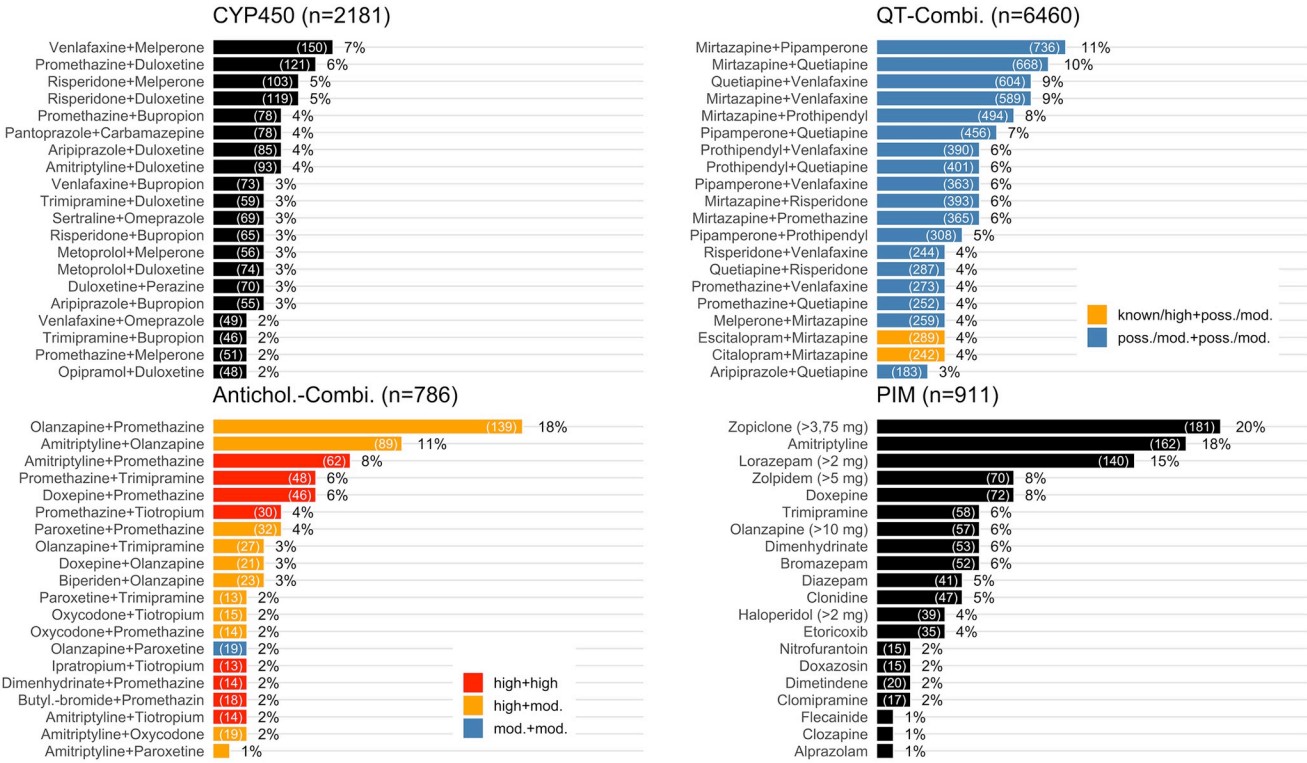

**Fig 3. TOP-20 drugs and drug combinations ranked by proportion of affected cases in total number of cases.** Double counting is possible, i.e. a case can have received several drugs or combinations of drugs. CYP450-Interaction: Pharmacokinetic cytochrome P450 (CYP)-mediated drug-drug interaction. QT-Combi.: A combination of at least two drugs on the same day with known/high or possible/moderate risk of TdP. Antichol. Combi.: A combination of at least two drugs on the same day with at least moderate anticholinergic activity. PIM: Potentially inappropriate medication in the elderly. Poss: Possible. Mod: Moderate.

strategies for treatment exist [50], such as cognitive-behavioral psychotherapy [51], or in severe cases electroconvulsive therapy [52] and vagus nerve stimulation [53].

Almost one half of all cases received a combination of multiple antidepressant drugs or a treatment with antidepressant drugs augmented by antipsychotic drugs. Several studies suggested that combining multiple antidepressants after failed monotherapy can be effective [54]. Our study found that tetracyclic antidepressants, in our study only mirtazapine, were the drug class used most frequently for combination therapy with SSRI, which is supported by current guidelines [5]. Moreover, we found relatively frequent combinations of mirtazapine with SNRI, which were shown to be effective by several studies but are currently not included in clinical guidelines [55–59]. The combination of multiple antidepressants could be advantageous in comparison to switching antidepressants as there is no need for titration and initial improvements might be maintained [60]. However, a combination strategy requires more attention to potential pDDI and PIM.

About 22% of patients received treatment with antidepressants augmented by antipsychotic drugs. The most frequent augmentation was conducted with atypical antipsychotics. Atypical antipsychotics were found to be effective as augmentation in major depressive disorders but also associated with an increased risk of discontinuation due to adverse events [61]. Therefore, current guidelines for the treatment of unipolar depression recommend augmentation therapy with the atypical antipsychotics only if previous monotherapy with antidepressant drugs failed [5]. However, not all antipsychotics were necessarily administered to address depressive

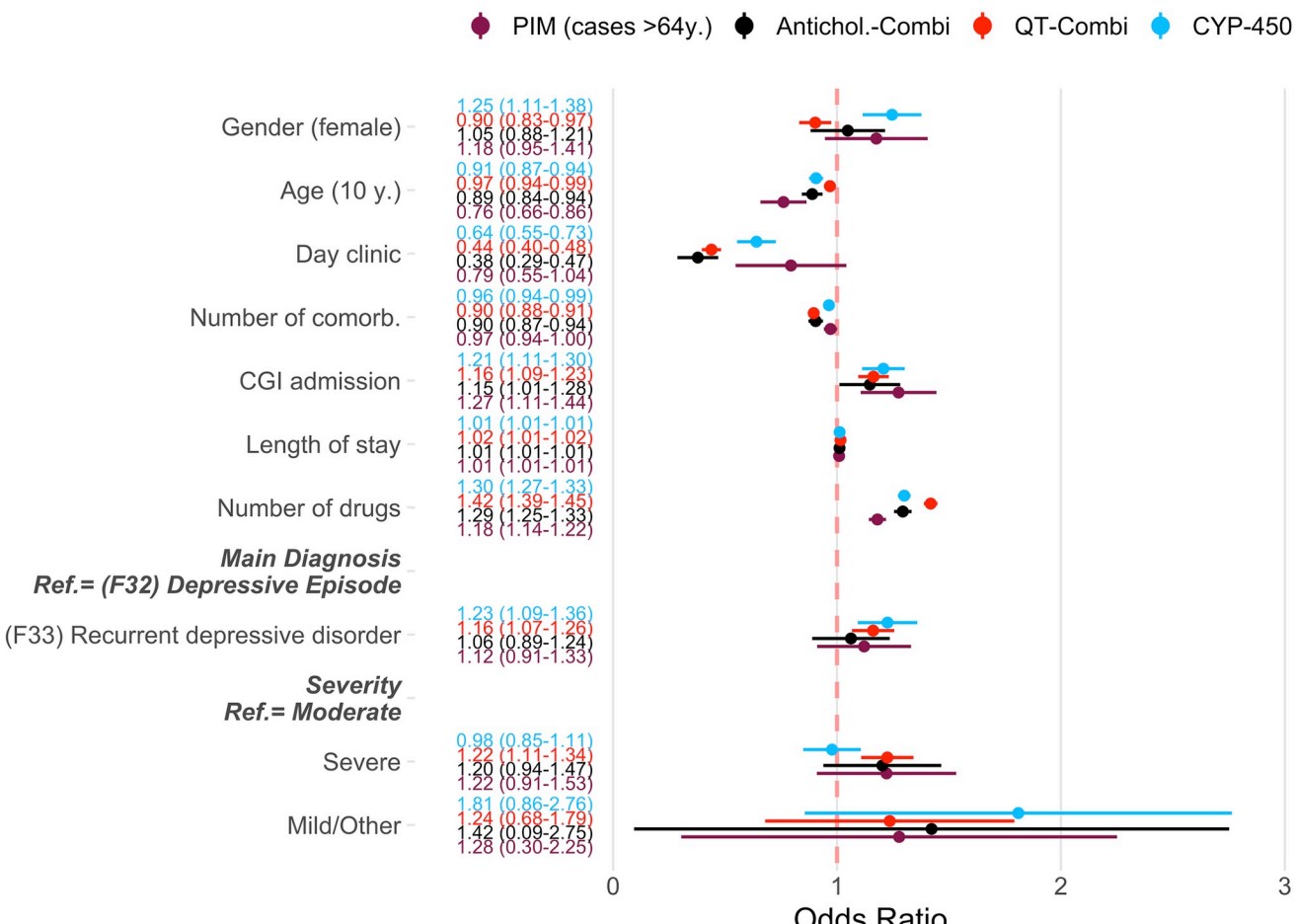

**Fig 4. Patient-specific risk factors for pDDI and PIM.** Circles show odds ratios of a multivariate logistic regression. These ratios reflect the multiplicative effect per influencing variable, i.e., for example, the risk of receiving at least one PIM increased by 18% for each additional drug taken, controlled for the other variables in the model. The values in brackets show the 95% confidence interval. Confidence intervals that do not include 1 show a statistically significant effect. CYP450-Interaction: Pharmacokinetic cytochrome P450 (CYP)-mediated drug-drug interaction. QT-Combi.: A combination of at least two drugs on the same day with known or possible risk of TdP. Antichol. Combi.: A combination of at least two drugs on the same day with at least moderate anticholinergic activity. PIM: Potentially inappropriate medication in the elderly. Mod.: Moderate.

disorders. Instead, they might have been used to address insomnia [62,63], control agitation and aggressive behaviour [64] or to treat comorbid psychotic symptoms. Augmentation therapy was identified to be superior to monotherapy with either antidepressant or antipsychotic drugs and to placebo in the acute treatment of psychotic depression [65,66], which accounted for 7.4% of our total sample (see Table 1). Furthermore, Lithium is recommended for augmentation therapy in unipolar depression by current guidelines [5]. Its relevant effects include recurrence prevention, acute-antidepressant effects and anti-suicidal effects. However, Lithium was used less frequently than other drugs in our study (2.8% of both total patient days and hospital cases).

We identified relatively few cases with a switch from one antidepressant drug to another. This result is in agreement with current studies that found that, following non-efficacy with an initial SSRI, only about one fifth of hospital cases remit and more than a half do not show a substantial benefit after a second-step switch to another monoaminergic

antidepressant drug [67]. Furthermore, three reviews illustrated that the effectiveness of switching between drug classes does not significantly differ from the switch within drug classes [68–70].

## Clinical implications with regard to pDDI and PIM

Combination and augmentation strategies increase the risk for pDDI and PIM. pDDI and PIM do not necessarily lead to ADR and negative patient outcomes. However, the association between undesired pDDI and PIM and an increased risk of negative patient outcomes has been firmly established by previous studies [8,16,17,71–75]. Indeed, the association between pDDI and actual negative patient outcomes might often be underestimated in psychiatry and therefore neglected in clinical practice [76].

Negative outcomes can be caused, for instance, by reduced or increased drug serum or plasma concentration, e.g. leading to the loss of desired drug effects or stronger ADR, respectively. In clinical practice, increased awareness of potential sources of pDDI and PIM can help medical staff to achieve desired and avoid undesired therapeutic outcomes [77]. Patients with multiple simultaneously administered drugs and patients requiring drugs with a narrow therapeutic range warrant close therapeutic drug monitoring to hold the patients' exposure within the desired therapeutic window [78]. The present study has quantified the problem, added an overview of the relevant risk factors and listed the main drugs and combination drug therapies accounting for potential pDDI and PIM in hospital psychiatry.

## The present study in comparison to previous research

The comparison of the present results to previous research was limited by scarce data considering pharmacoepidemiology of psychiatric hospital care for depressive disorders. The present study discerned most inpatients with depressive disorders to receive multiple psychotropic and non-psychotropic drugs. Rhee and Rosenheck found that 58% of depressed adults in their sample from office-based psychiatric care received more than one psychotropic drug simultaneously [79]. This number was slightly lower than the number in the present total sample (65%, not shown in Figures). However, in addition to the different settings, i.e. office-based versus inpatient treatment, a further explanation might be that Rhee and Rosenheck only considered the first 8 items per prescription for their analysis, while in the present study all pharmaceuticals were included.

Cascade et al. investigated the prevalence of antidepressant monotherapy versus combinations and found that 85% of patients treated by office-based physicians received antidepressant monotherapy [80]. This is contrast to the results of our study, which found that almost half of all cases received either a combination of multiple antidepressants or an augmentation with antipsychotics. However, the study by Cascade et al was carried out at office-based physicians and the prevalence of monotherapy decreased with increased severity of disease. Furthermore, psychiatrists (32%) were more likely to use a combination of multiple antidepressant drugs than primary care physicians (8%) and this is more similar to the percentage of cases that received an antidepressant combination therapy in the present study (25%). Augmentation therapy with antipsychotics was administered in only 2% of regimens in the study of Cascade et al published in 2007, which was far less than in the present study. However, Lenderts et al investigated the trends in pharmacotherapy of depressive disorders in 2009 after the approval of an atypical antipsychotic as augmentation to antidepressants by the FDA and their results showed a strong increase in augmentation therapy after the approval [81].

### Strength and weaknesses of the present study

A strength of this study is the profound and extensive set of daily medication data. Hence, a relatively detailed analysis was possible. Furthermore, a comparatively large number of patients from eight hospitals in Hesse, Germany, was included, providing a relatively representative picture of hospital care for depressive disorders in Germany.

The present study did not delineate patient-specific benefit-risk balances of prescriptions, for instance by including drug serum levels, results of electrocardiograms or individual pharmacogenetic risk factors. Therefore, it was not possible to differentiate between pDDI and actually inadequate prescriptions. Neither did our study document actual ADR related to pDDI, which would have required an entirely different study design and setting. However, the association between pDDI and the risk of ADR is well established by several previous studies [8,16,17,71–75].

## Conclusion

Most inpatients with depressive disorders receive multiple psychotropic and non-psychotropic drugs, and pDDI and PIM are relatively frequent. Few drugs accounted for a large fraction of cases. Due to the high prevalence and the potentially negative outcomes, patients taking a high number of different drugs require an intensive management of their individual drug-related risk-benefit profiles.

## Author Contributions

**Conceptualization:** Jan Wolff, Pamela Reißner, Gudrun Hefner, Claus Normann, Klaus Kaier, Harald Binder, Christoph Hiemke, Sermin Toto, Katharina Domschke.

**Data curation:** Jan Wolff, Gudrun Hefner, Ansgar Klimke.

**Formal analysis:** Jan Wolff, Gudrun Hefner, Klaus Kaier, Harald Binder, Christoph Hiemke, Katharina Domschke, Michael Marschollek, Ansgar Klimke.

**Funding acquisition:** Jan Wolff, Klaus Kaier, Harald Binder, Katharina Domschke, Ansgar Klimke.

**Investigation:** Jan Wolff, Pamela Reißner, Gudrun Hefner, Claus Normann, Sermin Toto, Katharina Domschke, Michael Marschollek, Ansgar Klimke.

**Methodology:** Jan Wolff, Pamela Reißner, Gudrun Hefner, Claus Normann, Klaus Kaier, Harald Binder, Christoph Hiemke, Sermin Toto, Katharina Domschke, Michael Marschollek, Ansgar Klimke.

**Project administration:** Jan Wolff, Pamela Reißner, Harald Binder, Katharina Domschke, Ansgar Klimke.

**Resources:** Jan Wolff, Ansgar Klimke.

**Software:** Jan Wolff, Ansgar Klimke.

**Supervision:** Jan Wolff, Claus Normann, Christoph Hiemke, Sermin Toto, Katharina Domschke, Michael Marschollek, Ansgar Klimke.

**Validation:** Jan Wolff, Pamela Reißner, Gudrun Hefner, Claus Normann, Klaus Kaier, Harald Binder, Christoph Hiemke, Sermin Toto, Katharina Domschke, Michael Marschollek, Ansgar Klimke.

**Visualization:** Jan Wolff.

**Writing – original draft:** Jan Wolff.

**Writing – review & editing:** Jan Wolff, Pamela Reißner, Gudrun Hefner, Claus Normann, Klaus Kaier, Harald Binder, Christoph Hiemke, Sermin Toto, Katharina Domschke, Michael Marschollek, Ansgar Klimke.

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
