## [Decision Letter · Decision Letter 0]

5 Jun 2021

PONE-D-21-11326

Pharmacotherapy, drug-drug interactions and potentially inappropriate medication in depressive disorders

PLOS ONE

Dear Dr. Wolff,

Thank you for submitting your manuscript to PLOS ONE. After careful consideration, we feel that it has merit but does not fully meet PLOS ONE’s publication criteria as it currently stands. Therefore, we invite you to submit a revised version of the manuscript that addresses the points raised during the review process.

We look forward to receiving your revised manuscript.

Kind regards,

Angela Lupattelli, PhD

Academic Editor

PLOS ONE

Journal Requirements:

Independent of the present study, KD received fees from Janssen Pharmaceuticals, Inc. for her consultancy work on the Neuroscience Steering Committee. CN received lecture and consultancy fees from Janssen-Cilag and Neuraxpharm as well as fees for conducting clinical studies from Janssen-Cilag. ST has received lecture fees from Janssen-Cilag, Otsuka / Lundbeck and Servier and is a member of the Advisory Board of Otsuka and Janssen-Cilag. CH has received lecture fees from Otsuka.

We note that you received funding from a commercial source: Janssen Pharmaceuticals, Inc, Janssen-Cilag and Neuraxpharm, Janssen-Cilag

5. Please include a copy of Table 1 which you refer to in your text.

Reviewers' comments:

Reviewer's Responses to Questions

**Comments to the Author**

1. Is the manuscript technically sound, and do the data support the conclusions?

Reviewer #1: Yes

Reviewer #2: Partly

2. Has the statistical analysis been performed appropriately and rigorously? 

Reviewer #1: Yes

Reviewer #2: Yes

3. Have the authors made all data underlying the findings in their manuscript fully available?

Reviewer #1: Yes

Reviewer #2: No

4. Is the manuscript presented in an intelligible fashion and written in standard English?

Reviewer #1: No

Reviewer #2: Yes

5. Review Comments to the Author

Reviewer #1: The paper is interesting and well written, but some critical issues are present.

-the study doesn't analyze the actual consequences of drug interactions. For this reason, the word "potential DDI" should be used instead of DDI, particularly in the results and discussion sections.

-some figures are difficult to understand, and the figure's legends are not clear. I suggest avoiding the insert of data or combination of data that are not relevant to the discussion

-in table 1, the variable "Length of stay" is showed with a range and not with a standard deviation

-in figure 1 is not clear the meaning of the gray bars. Moreover, the differences between groups seem not statistically significant. The figure could be deleted

-the drug classes reported in figure 2 are almost the same in the two groups. This figure could also be deleted describing the most important results in the text

-Figure 3 is not easy to read and understand immediately. However, I like it. In part 3A I cannot understand the meaning of the words "at intersection" (number of hospital days?). I suggest using in the figure and the whole text the term Combination AD/AP instead of Augmentation and the term Mono AP instead of Only AP. In figure 3C the sum of the percentages is 100% but the group Combination AP is missing.

-Figure 4 is not easily readable. Data should be presented as a table grouping data according to the number of drugs per day (e.g., 1-3, 4-5. 6-8, 9-11, >12)

-the description of Figure 5 in the results should be changed. The description should list the most involved drugs in each group (e.g. mirtazapine in QT-Combi). The suggestion to avoid certain drugs to substantially reduce the cases should be moved in the discussion with specific reference to the involved drugs.

Reviewer #2: OVERALL

The authors have conducted a very thorough descriptive analysis of the types of medications inpatients with major depressive disorders are taking during their hospital stay. However, without a more clear organization around specific aims being examined it is difficult to identify and interpret the most relevant findings from all of the data presented in the tables. This manuscript would benefit from a more focused set of aims build on a stronger justification for the clinical relevance of these analyses and a more concise presentation of data/results.

ABSTRACT: AIMS

1. The phrases “analyse prescription patterns and determinants of DDI and PMI” is vague. It is recommended that this be rephrased to provide more clarity of precisely what the authors intend to achieve with their analyses

INTRODUCTION

1. The introduction includes a lot of detail on the specifics of pharmacokinetic and pharmacodynamics of types of drugs without a clear and cohesive argument for how this is relevant to the aim of the study.

2. Potentially inappropriate medications are not clearly defined and a more cohesive argument is needed for why this is important and relevant to the aim of the study

3. More specificity with respect to “prescription patterns” and “determinants” is needed to understand what the aim of this manuscript and how it addresses concerns with DDI and PMI.

METHODS

1. This section would benefit from improved organization and greater detail about the study. Included subheadings such as Data source (i.e. the larger study), Study Sample (i.e. the sample used in the present analysis), Measures (i.e., the different types of drug categories and how they are identified, polypharmacy + definition, and patient characteristics and how they are measured/collected). Any measure reported in a table or results section should be described in the Methods/measurement section.

2. The authors state that this is part of a larger study and provided a reference for that study. However a brief 1-3 sentence description of the larger study would provide clarity to the context for the present study

3. It is not clear whether the sample for this study is a subsample from the larger study or uses the same sample but examines a different set of aims than the larger study. More detail about how the present study sample is derived from the larger study is needed

4. The paragraph “The study investigated . . .” seems to state three aims, which are different from the Aims stated at the end of the introduction. Aims should be stated at the end of the Introduction section and the Methods section should be used to described how these aims were examined.

5. More specificity is needed in the analysis section. There should be a clear link between each stated aim in the introduction, how this aim was assessed in the analysis section. For example, in the logistic regression model, what is the outcome? What is the main predictor of interest? What other covariates are being adjusted for as potential confounders of the relationship you are aiming to examine?

RESULTS

1. The author present a lot of very detail results and descriptive tables. However, there is not a clear connection between the data being presented in the tables and figures and how the results are relevant to the aims of the study. Without a clear connection between the Aims, analytic methods, and results presented in the Tables and Figures it is difficult to identify the most relevant findings with respect to the aims of the manuscript. It is recommended that the authors reduce the amount of data presented in terms of both the number of tables and figures and the contents of those table to include only data relevant to a narrower, more focused set of aims.

2. It is not clear why the authors choose to make a distinction between “recurrent depressive disorders” and “depressive episodes”. Are they authors hypothesizing that DDI or PMI would be more prevalent in one of these types of depression? If so, this should be stated in the aims with a justification in the background section.

3. Similarly, in Figure 1, the mild/other, moderate, and severe stratifications are not defined and the connection between the data presented and the aims of the study are not clear.

4. There are numerous measure in Table 1 that are not described in the Methods section (e.g. Day-clinic, number of comorbidities). The methods section should make clear how/where all data was obtained and how any summary measures were created. For example, what conditions were counted in the “comorbidities” variable. Additionally, the relevance of these measure to the aims should be clarified. Are they being examined as predictors, potential confounders, or outcomes of interest in the multivariate models?

5. The terms “combination”, “augmenting”, and “switching” are introduced in Figure 3 without being defined in the methods section. Similar to comments above, it is not clear how descriptions of “combing”, “augmenting”, “switching”, etc. relate to DDI or PMI.

DISCUSSION

1. The discussion include a very thorough review of guidelines and approaches to pharmacotherapy for depression and quantify the prevalence of each treatment approach (e.g., monotherapy, ADs augmented with antipsychotics). However, the clinical value of this information are not clear. The Discussion would benefit from greater organization focused around the key findings.

2. It seems logical that patients with more severe or recurrent depression would be more likely to be treated with multiple medications. Additionally, comorbidity of psychiatric conditions (and other health conditions), including serious mental illnesses with psychotic features, may warrant use of antipsychotic and mood stabilizing medications. Possible comorbidities were not included in the analyses or discussed as a limitation.

TABLES and FIGURES

A lot of very detailed information is presented and it is difficult to identify what pieces of information are most important/relevant. It is recommended that the tables and figures be reduced in terms of total number and content to focus on the most important outcomes of interest.

6. PLOS authors have the option to publish the peer review history of their article (what does this mean?). If published, this will include your full peer review and any attached files.

Reviewer #1: **Yes: **prof. Roberto Leone

Reviewer #2: No

---

## [Author Response · Author response to Decision Letter 0]

7 Jul 2021

Please see the attached PDF "Answers to reviewer comments".

---

## [Editor Report · Decision Letter 1]

12 Jul 2021

Pharmacotherapy, drug-drug interactions and potentially inappropriate medication in depressive disorders

PONE-D-21-11326R1

Dear Dr. Wolff,

We’re pleased to inform you that your manuscript has been judged scientifically suitable for publication and will be formally accepted for publication once it meets all outstanding technical requirements.

Kind regards,

Angela Lupattelli, PhD

Academic Editor

PLOS ONE

---

## [Editor Report · Acceptance letter]

14 Jul 2021

PONE-D-21-11326R1 

Pharmacotherapy, drug-drug interactions and potentially inappropriate medication in depressive disorders 

Dear Dr. Wolff:

I'm pleased to inform you that your manuscript has been deemed suitable for publication in PLOS ONE. Congratulations! Your manuscript is now with our production department. 

Kind regards, 

on behalf of

Dr. Angela Lupattelli 

Academic Editor

PLOS ONE